# Inferring from Logits: Exploring Best Practices for Decoding-Free Generative Candidate Selection

## Abstract

Generative Language Models rely on autoregressive decoding to produce the output sequence token by token. Some tasks, such as preference optimization, require the model to produce task-level output consisting of multiple tokens directly by selecting candidates from a pool as predictions. Determining a task-level prediction from candidates using the ordinary token-level decoding mechanism is constrained by time-consuming decoding and interrupted gradients by discrete token selection. Existing works have been using decoding-free candidate selection methods to obtain candidate probability from initial output logits over vocabulary. Though these estimation methods are widely used, they are not systematically evaluated, especially on end tasks. We introduce an evaluation of a comprehensive collection of decoding-free candidate selection approaches on a comprehensive set of tasks, including five multiple-choice QA tasks with a small candidate pool and four clinical decision tasks with a massive amount of candidates, some with 10k+ options. We evaluate the estimation methods paired with a wide spectrum of foundation LMs covering different architectures, sizes and training paradigms. The results and insights from our analysis could inform the future model design.

## 1 Introduction

Large Language Models (LLMs) have shown amazing performance after pre-training on a massive corpus (Lewis et al., 2020), instruction tuning (Longpre et al., 2023), and preference alignment (Ethayarajh et al., 2024). Generative LMs respond to queries by generating tokens to form an output sequence and optimize themselves by learning to generate the correct tokens. The simplicity of token-level inference and optimization compromises its performance on end tasks, as there is a gap between the token-level paradigm and sequence-level task results and learning signals.

Some tasks use generative LM to select the answer(s) from a given pool of options where each candidate answer is a natural language sequence. With such a task formulation, both LLMs' generalizable reasoning capabilities on novel scenarios and domain expertise contained in the candidate space can be utilized. Multiple-choice QA considers answer options as the candidate pool (Khashabi et al., 2020). The large collection of labels are candidate answers for extreme label classification tasks (Amigo & Delgado, 2022). The retrieval task aims to extract relevant documents from a large-scale evidence corpus (Lu et al., 2023). The candidate pool can also be expert-curated professional coding systems (Taylor et al., 2022). For example, when instructing the model to prescribe medications, items in drug databases are candidates (Fleming et al., 2023); when conducting diagnoses, disease ontology forms the candidate space (Singhal et al., 2023). The typical practice is to use full decoding to generate an output sequence and then match the natural language output with candidates in the pool to select candidates to be predicted as answers (Mishra et al., 2022). However, selecting candidates using full decoding not only cuts off the gradient flow and disables direct optimization on decoded results but also limits the output bandwidth due to time-consuming discrete decoding.

Existing works perform candidate selection without decoding for outcome-level optimization or efficient parallel predictions. For example, Ma et al. (2023b) calculate averaged logits of MCQA options to select an answer without decoding; Xu et al. (2023b) estimate the NLI result using logits of a single token. Though these decoding-free candidate selection practices are widely used, there is no formal definition or clear investigation of the properties of each method. There is also no consensus

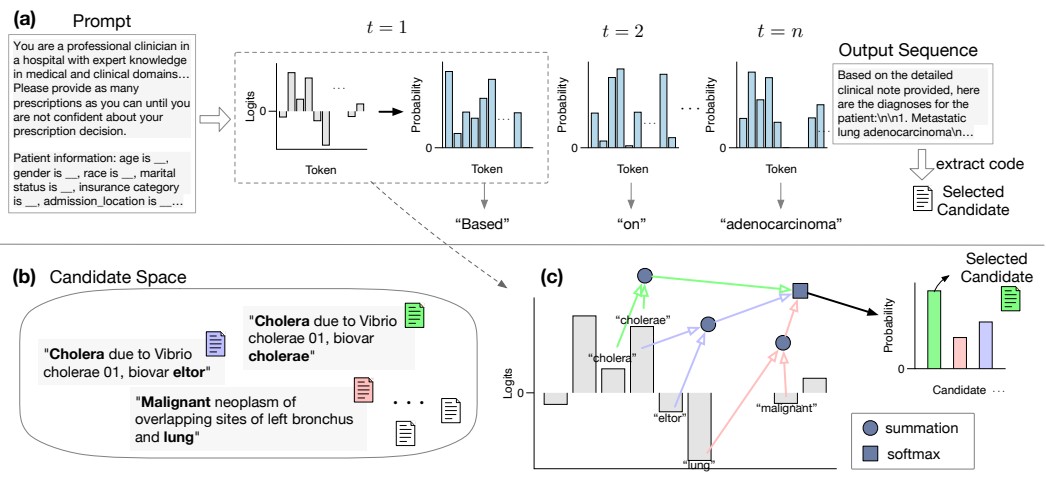

Figure 1: (a) Using full decoding for tasks with answer candidates by generating output sequences token-by-token. The task is to identify diagnoses given the patient's medical record. (b) Candidate space, consisting of coded diagnoses. (c) Using decoding-free generative candidate selection method for the diagnoses task based on prior-decoding logits over vocabulary.

about the guiding principles for deploying those methods under various tasks and data scenarios with diverse numbers, lengths, and complexity of candidate sequences. In this work, we formally define the decoding-free generative candidate selection task, and conduct the first systematic evaluation of typical practices on downstream tasks, reflecting the ultimate influence of the selection methods compared with conducting full decoding. Our systematic evaluation covers an **extensive collection of candidate selection methods**, including five decoding-free approaches to calculating candidate probability distribution from token logits, as well as encoding-only dense retrieval method and full decoding approach. The effects of deploying various methods are evaluated with a **comprehensive set of downstream testbeds** widely used for LLM evaluation. The first type of testbed includes five multiple-choice QA tasks with broad target capabilities and candidate diversity, reflecting the candidate selection capabilities while the candidate pool is limited. We further increase the difficulty and examine the performance on tasks with massive numbers of candidates on expert-curated large ontology with 10k+ options for making diagnoses, procedure decisions, ordering lab tests, and prescribing medications. Finally, we dive into the characteristic shifts of candidate selection methods while using a **wide spectrum of foundational LMs**. The base models are diverse in terms of architectures (decoder-only or encoder-decoder), sizes (spanning from 137M to 11B), and training methods (pre-trained or instruction-tuned).

The evaluation provides insights into the properties of decoding-free candidate selection methods. The performance of the token-logits-based candidate representation is highly dependent on the properties of the pretrained LM, dataset domain difficulty, and candidate space diversity. Pure estimation methods can outperform non-instruction-tuned models due to the challenges faced by weak base models in handling certain question formats during decoding. In this case, estimation methods offer a more straightforward means of exhibiting knowledge through token logits. The insights derived from our evaluation enable more informed and confident design for future estimation methods. We empirically demonstrate that the logits of the first output step are most informative; using selective tokens for estimations compromises the performance and scaling properties of various model sizes.

## 2 PROBLEM FORMULATION

### 2.1 DECODING AND TRAINING PARADIGM OF GENERATIVE LMs

The ordinary sequence-to-sequence formulation of generative LM takes the input sequence $seq_{in} = t_1^{in}, \ldots, t_{|seq_{in}|}^{in}$ and is expected to generate an output sequence $\hat{seq}_{out} = \hat{t_1^{out}}, \ldots, \hat{t_{|seq_{out}|}^{out}}$. The output sequence generation involves encoding the input sequence to contextual vector representation (*i.e.*, output of the final transformer block), and decoding the outputs following

$$\hat{seq}_{out} = f_{\text{full-decode}}(f_{\text{encode}}(seq_{in})). \tag{1}$$

During inference, the decoding function $f_{\text{full-decode}}$ involves $|seq_{out}|$ discrete decoding steps, in which each step produces one output token. For $(k+1)-th$ step of decoding, which is conditioned on both the input sequence and $k$ generated tokens, the model produces logits $\mathbf{z}_k \in \mathbb{R}^{|V|}$ over the vocabulary $V$ after passing output of encoding through unembedding matrix, and then obtain a probability distribution over the possible next token ($w \in V$) in the output sequence $P\left(w \mid t^{in}_{1:|seq_{in}|}, \hat{t}^{out}_{1:k}\right) = \text{softmax}(\mathbf{z})$. Then a discrete token at this autoregressive decoding step is produced by Equation 2.

During training, The LM is trained to minimize the difference between the generated tokens with tokens in the ground-truth output $seq_{out}$. The LM is optimized to minimize the cross-entropy loss shown in Equation 3 applied on the probability of the *gold* next token conditioned on the *gold* target output tokens in the previous segment in a teacher-forcing manner, assuming the $|seq_{out}|$-th token marks the end of the sentence.

$$\hat{t}^{out}_{k+1} = \text{argmax}_{w \in V} \, P\left(w \mid t^{in}_{1:|seq_{in}|}, \hat{t}^{out}_{1:k}\right) \quad (2) \qquad \mathcal{L}_{CE} = \sum_{k=0}^{|seq_{out}|} -\log P\left(t^{out}_{k+1} \mid t^{in}_{1:|seq_{in}|}, t^{out}_{1:k}\right) \quad (3)$$

## 2.2 Candidate selection with a candidate answer pool

When performing tasks using generative LMs, we include task instruction and query in the input sequence $seq_{in}$ and expect the derived answer of the query $\hat{ans}$ from the generated output sequence $\hat{seq}_{out}$ can match the ground-truth answer $ans$. Some tasks expect open-ended free generation where the final answer is the generated output ($\hat{ans} = \hat{seq}_{out}$), such as translation, creative story generation and dialogue conversation (Wang et al., 2023; Ma et al., 2023a). However, many tasks have an existing answer candidate pool, and the output sequence needs to find a matched candidate as the final prediction. We notate the candidate pool as $C$ and a candidate as $c \in \{c_1, c_2, \ldots, c_{|C|}\}$ where $|C|$ is the total number of candidate options. Each candidate is a natural language sequence. The answer of the query has to be one of the candidate, *i.e.* $ans \in C$. For example, answer options are candidates for multiple-choice question answering (Talmor et al., 2018), segments of the input sentences are candidates of information extraction (Sun et al., 2024; Zhao et al., 2024), passages in large archives serve as candidates for information retrieval (Lewis et al., 2021), and drugs within medication databases are candidates for prescription tasks (Yu et al., 2024). Below are two examples from a common sense multiple-choice question dataset:

**Example 1.** *Question:* *The fox walked from the city into the forest, what was it looking for?* *Candidates:* *pretty flowers; natural habitat; storybook; dense forest* **Answer:** *natural habitat*

## 2.3 Ordinary approaches for candidate selection

**Classification methods** train a $|C|$-way classification head with specialized parameters, where each candidate is treated as a class label. **Retrieval methods** first create an index with an encoded representation of each candidate. During prediction, the most matched candidates are retrieved, where the match is measured by the similarity between the candidate embedding and the query.

To select candidates using a **generative approach with full decoding**, the LM first generates a free-form output with discrete tokens $\hat{seq}_{out}$ by decoding the most probable words at each output step, then an additional mapping $f_{\text{map}}$ from $\hat{seq}_{out}$ to candidates is needed to produce the probability over all candidates following Equation 4. This function can be a heuristic rule, semantic similarity matching, or manual processing.

$$P\left(c \mid t^{in}_{1:|seq_{in}|}\right) = \{p_{c^1}, p_{c^2}, \ldots, p_{c^{|C|}}\} = f_{\text{map}}(f_{\text{full-decode}}(f_{\text{encode}}(seq_{in}))) \quad (4)$$

Then, the predicted answer $\hat{ans}$ where $\hat{ans} \in C$ is produced by $\hat{ans} = \text{argmax}_{c \in C} \, P\left(c \mid t^{in}_{1:|seq_{in}|}\right)$.

## 2.4 Generative candidate selection without decoding

At this point, we formally define the task of **decoding-free generative candidate selection**. In §2.3, we introduce *generative candidate selection using decoding*, where the answer is reflected by the output sequences. However, there are multiple severe limitations of decoding-based candidate selection. On the one hand, the discrete argmax operator for token selection interrupts the gradient flow, making applying objectives on task outcomes inefficient, such as using reinforcement learning with one-per-outcome sparse rewards instead of token-level feedback. On the other hand, the decoding process is time and resource-consuming, limiting the output bandwidth of generative LMs.

*Decoding-free generative candidate selection* $f_{est}$ is a function to produce the candidate prediction probability given $seq_{in}$ without discrete decoding. Given the encoded representation, the function calculates the logits of the first decoding step $\mathbf{z}_0$ (before the first discrete decoding) and then performs various approaches on top of the logits to estimate the probability of candidate outcomes. The estimation is driven by the token logits information under the assumption that the model's intended preference over outcome candidates can be reflected in logits of tokens of the candidate sequences.

The estimation method directly produces the probability distribution over all candidate outputs following Equation 5. The predicted answer can be yielded by $a\hat{n}s = \text{argmax}_{c \in C} P\left(c \mid t^{in}_{1:|seq_{in}|}\right)$.

$$P\left(c \mid t^{in}_{1:|seq_{in}|}\right) = f_{est}(f_{encode}(seq_{in})) \tag{5}$$

Note that the full decoding process $f_{full-decode}$, including probability calculation, argmax for token selection, decoding for the next token, etc, is not conducted. The mapping from discrete output sequence to candidates $f_{map}$ is not needed either.

Decoding-free candidate selection methods are especially beneficial for two scenarios. 1) *Accurate outcome-based optimization.* To optimize the model with the feedback directly from the predicted outcome $a\hat{n}s$, we need to know the model's prediction over potential candidates $C$ without interrupting the gradient flow. 2) *Efficient answer production.* The token dependency of full decoding prevents the decoding mechanism from outputting the answer in parallel. Even though the output sequence $se\hat{q}_{out}$ is generated, an additional step ($f_{map}$) is needed to convert the output sequence $se\hat{q}_{out}$ to the predicted answer $a\hat{n}s$. Decoding-free candidate selection produces the probability over all potential answers directly without autoregressive generation and supports parallel inference, significantly improving the time and resources needed for producing the answers to queries.

### 2.5 DIFFERENCE COMPARED WITH ORDINARY APPROACHES

The *common property* between the generative candidate selection and classification is that both settings require a given set of selections to produce the final output. However, candidate selection is *different* from classification in many key aspects summarized in Table 1, specifically: 1) *Support of dynamic candidates.* A classification model has to use the same set of output labels across all instances (*e.g.* positive or negative for sentiment classification). However, candidate selection methods allow the task to have a different set of output candidates for each instance (*e.g.* different answer options for each question in MCQA). 2) *No need of additional parameters.* The classification head is an additional set of parameters specialized for the defined output classes. Different classification tasks have to use a separate set of parameters. Candidate selection methods support various candidates with the generative LM's native parameters only, without any additional parameters. 3) *No need for specialized training.* The additional parameters for each classification task need to be parameterized by training on task-specific data, preventing it from generalizing to new tasks or labels. However, decoding-free estimation methods support zero-shot candidate selection, as the possibilities of choosing options are calculated dynamically according to corresponding candidates.

Table 1: Key difference between classification and generative candidate selection.

| Property | Classification | Generative candidate selection |
|---|---|---|
| Candidate pool | Fixed for all instances | Dynamic for each instance |
| Separate parameters | Need separate classification head | No separate parameters |
| Specialized training | Require training | Support on the fly zero-shot inference |

## 3 GENERATIVE CANDIDATE SELECTION METHODS

The decoding-free generative candidate selection methods (also referred to as estimation methods) produce the probabilities of predicting each candidate using the logits obtained at the first output step. We introduce several decoding-free generative candidate selection methods, including $k$-th token, token average and token sum, as well as other candidate selection methods for comparison, including full decoding and dense retrieval. We also investigate other interesting factors that influence the estimation performance, including the *candidate sequence keyword selection* deciding which tokens are considered to represent a candidate option, and the *output step used to obtain the logits* determining the source of raw logits data. We discuss these factors in §5.2 and §5.3. Though

some of the designs have been used by existing works, there is no justification or empirical analysis to support their design choices. To the best of our knowledge, this work is the first to provide a formal summary of these approaches and systematically investigate the properties of each design of generative candidate selection methods.

### 3.1 ESTIMATION CANDIDATE PROBABILITIES FROM LOGITS

**Estimation by logits of $k$-th token.** From the logits across all tokens in the vocabulary $\mathbf{z}_0$, we calculate the logit for a single token (*e.g.*, the first or the last token) of each candidate sequence and apply softmax to these selected logits to determine the probability $p_{c^i}$ of predicting a candidate $c^i$ among all candidates $C$ following Equation 6. The candidate is represented by a sequence containing $|c^i|$ tokens, *i.e.*, $c^i = c^i_1, \ldots, c^i_{|c^i|}$. For instance, in Example 1, logits for "pretty", "natural", "storybook" and "forest" can be extracted to calculate the probability of each choice based on the first token. We consider two variants in our evaluation: *first token estimation* and *last token estimation*.

**Estimation by averaged token logits.** We average the logits across all tokens for each candidate and apply softmax to these averaged logits across all candidates to compute choice probabilities following Equation 7 where $\langle \cdot \rangle$ represents the averaging operator. For "pretty flowers" choice in Example 1, the logits on "pretty" and "flowser" are selected from $\mathbf{z}_0$. The selected logits are averaged across tokens in the candidate sequence, and then softmax is applied across candidates.

**Estimation by sum of token logits.** For each choice, we sum the logits across all tokens of the candidate sequence. We then apply softmax to these summed logits to determine the probability of selecting each choice following Equation 8. In this approach, choices with more tokens tend to have a higher probability of selection.

$$p_{c^i} = \frac{\exp(logit(c^i_k))}{\sum_{j=0}^{|C|} \exp(logit(c^j_k))} \qquad (6) \qquad p_{c^i} = \frac{\exp(\langle logit(c^i) \rangle)}{\sum_{j=0}^{|C|} \exp(\langle logit(c^j) \rangle)} \qquad (7) \qquad p_{c^i} = \frac{\exp(\sum_{k=0}^{|c^i|} logit(c^i_k))}{\sum_{j=0}^{|C|} \exp(\sum_{k=0}^{|c^j|} logit(c^j_k))} \qquad (8)$$

### 3.2 ORDINARY CANDIDATE SELECTION METHODS TO COMPARE

**Full decoding.** Following the decoding paradigm introduced in §2.1 and how to induce answers within a candidate pool from the full decoded output sequence introduced in §2.3, we perform full decoding to obtain an output sequence, then use a mapping function to find the corresponding predicted answer from the given candidate pool. The mapping function is task-dependent; we use the task-specific mapping rules introduced along with each dataset. Typical practices include using regular expressions to match patterns in the output sequences, such as phrases like "Answers: ", and predicting the candidate with the highest semantic similarity with the output sequence.

**Dense retrieval.** In the retrieval baseline, we formulate the candidate selection task as a retrieval task and use dense passage retrieval as one of the reference models. Separate encoders are employed for encoding queries and candidate choices. Specifically, the question and each candidate choice are embedded into a high-dimensional vector space using these encoders. Cosine similarity is then computed between the question embedding and each candidate choice embedding. This similarity score quantifies the relevance of each choice to the posed question and determines the probability of each choice being the correct answer. For our experimental setup, we use the Facebook DPR question encoder and context encoder (Karpukhin et al., 2020) to generate embeddings of the questions and candidate choices, respectively.

## 4 EVALUATION SETTINGS

To contextualize the real-world performance of various candidate selection methods, we apply the introduced methods to ultimate downstream tasks to reflect their influence on end tasks. We introduce the selected tasks in §4.1 and base generative LMs used in experiments in §4.2.

### 4.1 TESTBED TASKS FOR CANDIDATE SELECTION

We evaluate generative candidate selection methods on two typical types of candidate selection tasks. The first type contains a limited number of answer candidates so that all plausible choices can fit in the input prompt of the model if needed. The second type of task has a massive candidate pool with a

large amount of candidates, which cannot fit in the input prompt. We show a comparison between these two settings in Table 5. We cover more details of the testbed tasks in Appendix B.1 and the distribution of candidate sequences lengths in Appendix B.2.

### 4.1.1 TASKS WITH LIMITED NUMBERS OF CANDIDATES

We use five tasks with the provided candidate pools: (1) CommonsenseQA (Talmor et al., 2018), (2) MMLU (Hendrycks et al., 2021b;a), (3) GPQA (Rein et al., 2023), (4) BIG-Bench (Srivastava et al., 2022), and (5) ARC (Clark et al., 2018), covering commonsense questions, science and liberal arts subjects in different education levels, logical reasoning questions, etc. Instances in all datasets contain one correct option and multiple distractors. They vary in difficulty, candidate option lengths, and number of candidates per instance, as shown by data statistics in Table 2. We report accuracy and per-instance runtime for these tasks and select one of the dataset splits with available answer keys.

To require the model to answer in a specific format without intermediate thinking processes, we add specific instructions in the input prompt, as shown in Appendix B.3. When incorporating the candidate information, we use candidate sequences without indication heads (*e.g.* A, B) to estimate the selection for decoding-free methods. For the full decoding baseline, candidate sequences with indicators are included in the input for a fair comparison. For the mapping function $f_{\mathrm{map}}$ used by the full decoding, which converts output sequence $seq_{out}$ to candidate selection $a\hat{n}s$, we capture the first occurrence of a candidate sequence or indication head with regular expressions as the prediction as further elaborated in Appendix B.4.

Table 2: Properties of testbeds with the limited and massive number of candidates.

| Task | Split | Instances # | Candidate # | Avg. candidate token length |
|------|-------|-------------|-------------|----------------------------|
| CommonsenseQA | train | 9741 | 5 | 1.52 |
| MMLU | test | 14,042 | 4 | 6.72 |
| GPQA | train | 448 | 4 | 5.84 |
| BIG-Bench | train | 250 | 3 | 5.33 |
| ARC | test | 2,241 | 4 | 3.76 |
| Diagnoses | test | 1,081 | 94,739 | 9.65 |
| Procedures | test | 1,054 | 85,257 | 9.37 |
| Lab Orders | test | 1,067 | 1,622 | 5.25 |
| Prescriptions | test | 1,036 | 24,785 | 2.30 |

### 4.1.2 TASKS WITH MASSIVE NUMBERS OF CANDIDATES

We adapt four professional decision-making tasks introduced by Ma et al. (2024) where the answer has to fall in a large-scale expert-defined coding system as the second category testbeds. The goal is to select multiple candidates from the pool as the predicted clinical decisions. They include: **(6) Diagnosis decisions on ICD-10-CM coding system.** Given the patient records of a hospital admission and the history diagnoses of the patient, the task aims to produce a set of diagnoses, each has to choose from chapters in the International Classification of Diseases (10th revision) coding system with 94k+ options. **(7) Procedure decisions on ICD-10-PCS coding system.** The task determines a set of actions to be implemented to intervene in the patient's health status given patient record at admission time. Candidates for procedures are level 2 codes in ICD-10-Procedure Coding System ontology with 85k+ options. **(8) Lab orders on LOINC coding system.** Given the admission patient record, the task selects a set of lab items from the candidate pool of 3rd-level codes of the Logical Observation Identifiers Names and Codes system. **(9) Prescriptions on ATC coding system.** The goal is to identify a set of medications, each coded as a pharmacological subgroup in the Anatomical Therapeutic Chemical classification system, to be prescribed to the patient given admission medical record.

### 4.2 BASE GENERATIVE LMs.

We assess decoding-free candidate selection approaches while using various pretrained generative language models, including both decoder-only models in the Mistral and LLaMA families, as well as encoder-decoder models in the Flan-T5 family. For LLaMA (AI@Meta, 2024) and Mistral (Jiang et al., 2023) models, we use both models without instruction tuning (*LLaMA3 8B* and *Mistral v0.3*

Table 3: Accuracy and runtime per instance (in seconds) for each method across five multiple-choice QA datasets with a limited number of candidates per question. For generative candidate selection methods without decoding, we report the performance gap compared with full decoding. Methods that underperform or outperform full decoding are highlighted with red or green background.

| Model (# Param) | Method | Acc. Runtime CommonsenseQA | | Acc. Runtime MMLU | | Acc. Runtime GPQA | | Acc. Runtime BIG-Bench | | Acc. Runtime ARC | |
|---|---|---|---|---|---|---|---|---|---|---|---|
| LLaMA3 (8B) | Decoding | 31.83 | 0.69 | 36.53 | 0.84 | 27.90 | 0.45 | 34.00 | 0.27 | 55.51 | 1.12 |
| | First | +9.11 | 0.05 | -6.8 | 0.07 | -3.12 | 0.20 | -2 | 0.09 | -12.63 | 0.06 |
| | Last | +9.23 | 0.05 | -7.89 | 0.07 | -2.01 | 0.20 | -2 | 0.09 | -12.67 | 0.06 |
| | Average | +3.24 | 0.08 | -4.33 | 0.11 | -5.36 | 0.31 | -0 | 0.13 | -3.52 | 0.10 |
| | Sum | +4.81 | 0.08 | -3.75 | 0.11 | -5.58 | 0.31 | +0.8 | 0.13 | -8.88 | 0.10 |
| LLaMA3 Instruct (8B) | Decoding | 70.70 | 0.16 | 58.86 | 1.42 | 27.68 | 1.31 | 51.20 | 0.25 | 91.70 | 0.19 |
| | First | -38.34 | 0.04 | -31.83 | 0.06 | -2.68 | 0.19 | -19.2 | 0.08 | -54.08 | 0.05 |
| | Last | -38.16 | 0.04 | -32.78 | 0.06 | -3.13 | 0.19 | -19.2 | 0.08 | -56.72 | 0.05 |
| | Average | -36.08 | 0.06 | -32.55 | 0.10 | -7.14 | 0.29 | -15.6 | 0.12 | -52.74 | 0.08 |
| | Sum | -36.82 | 0.06 | -32.7 | 0.10 | -5.8 | 0.29 | -16 | 0.12 | -53.01 | 0.08 |
| Mistral v0.3 (7.3B) | Decoding | 21.63 | 1.28 | 25.51 | 0.96 | 30.13 | 0.75 | 30.00 | 0.90 | 29.27 | 1.24 |
| | First | +26.79 | 0.04 | +4.66 | 0.07 | -3.79 | 0.19 | +2.4 | 0.08 | +19.41 | 0.06 |
| | Last | +27.25 | 0.04 | +4.02 | 0.07 | -3.57 | 0.19 | +2 | 0.08 | +18.25 | 0.06 |
| | Average | +20.89 | 0.06 | +7.04 | 0.10 | -4.01 | 0.27 | +2 | 0.12 | +25.17 | 0.08 |
| | Sum | +25.16 | 0.06 | +7.62 | 0.10 | -5.58 | 0.27 | +1.2 | 0.12 | +24.05 | 0.08 |
| Mistral Instruct v0.3 (7.3B) | Decoding | 65.12 | 0.72 | 52.06 | 1.39 | 29.02 | 1.41 | 47.60 | 0.85 | 86.43 | 0.86 |
| | First | -18.34 | 0.04 | -21.52 | 0.06 | -4.02 | 0.19 | -16 | 0.08 | -34.13 | 0.05 |
| | Last | -18.52 | 0.04 | -22.76 | 0.06 | -2.46 | 0.19 | -15.6 | 0.08 | -35.92 | 0.05 |
| | Average | -20.74 | 0.05 | -19.48 | 0.09 | -4.91 | 0.26 | -16 | 0.11 | -27.08 | 0.07 |
| | Sum | -17.42 | 0.05 | -18.87 | 0.09 | -5.58 | 0.27 | -16.4 | 0.11 | -27.08 | 0.07 |
| Flan-T5-XL (11B) | Decoding | 97.48 | 0.27 | 48.36 | 0.27 | 25.45 | 0.29 | 65.20 | 0.35 | 89.25 | 0.29 |
| | First | -44.96 | 0.00 | -21.49 | 0.00 | -1.34 | 0.01 | -32.8 | 0.00 | -42.44 | 0.00 |
| | Last | -45.53 | 0.00 | -21.25 | 0.00 | -0.9 | 0.01 | -33.2 | 0.00 | -44.49 | 0.00 |
| | Average | -40.05 | 0.02 | -18.54 | 0.03 | -1.79 | 0.09 | -31.2 | 0.04 | -33.83 | 0.03 |
| | Sum | -46.84 | 0.02 | -21.47 | 0.03 | -4.02 | 0.09 | -32 | 0.04 | -39.9 | 0.03 |
| Facebook DPR | Retrieval | 32.07 | 0.25 | 27.15 | 1.31 | 25.22 | 0.13 | 30.80 | 0.01 | 39.76 | 0.10 |
| - | Random | 20.00 | 0.00 | 25.00 | 0.00 | 25.00 | 0.00 | 33.33 | 0.00 | 25.00 | 0.00 |

*7B*) and after instruction tuning (*LLaMA3 Instruct 8B* and *Mistral Instruct v0.3 7B*). Among Flan-T5 models, we use the 11B variant (Chung et al., 2022). When preparing the input sequence $seq_{in}$, we apply the chat template for the models trained with the prompt template, and we append the generation prompt to indicate the start of the answer segment. Additionally, we include a random guess baseline to represent the expected metrics achieved by chance.

## 5 EXPERIMENTS RESULTS

Table 3 shows the candidate selection performance on five MCQA tasks with limited candidates. We show the performance on four clinical tasks with large-scale candidate pools in Table 4. Given longer candidate sequences, we introduce a new decoding-free candidate selection approach named `Sample Avg.`, which calculates average logits for *every other* token in candidate sequences. Besides the analysis for output steps, candidate token selection (Figure 2), candidate length and model sizes (Figure 3), we additionally demonstrate that adding chat template for instruction-tuned model hurts the estimation performance in Appendix C.1, additional ablation study on candidate length in Appendix C.2 and performance breakdown in Appendix C.3.

### 5.1 CHARACTERISTICS OF GENERATIVE CANDIDATE SELECTION METHODS

**Insight 1: Estimation methods provide reasonable initial guesses for challenging tasks and decision intuition especially when full decoding is weak.** In Table 3 with limited candidates, for more challenging datasets such as GPQA, decoding-free candidate selection approaches (also referred to as "estimation methods") provide a reasonable initial guess and do not necessarily perform significantly worse than full decoding. Compared to full decoding, estimation methods even provide better performance for CommonsenseQA using LLaMA3 and all MCQA tasks except GPQA using Mistral v0.3. We observe these two models still struggle to handle the format for answering the question for some tasks during decoding, so it is hard to project its knowledge to interpretable results since the only surface to represent knowledge, outputting sequences, is not working for a weak base model. While knowledge by estimation methods is easier to exhibit through token logits.

For the results on clinical decisions with massive candidates presented in Table 4, all methods experience a decrease in performance on these more challenging tasks compared to the ones with a limited candidate space. Among decoder-only models, estimation methods can outperform full decoding for lab orders and prescriptions, particularly in non-instruction-tuned variants. Specifically, all estimation approaches surpass Mistral v0.3 in lab test orders, with `Sample Avg.` achieving the highest increase of 29.25 points compared to full decoding. Additionally, four of the estimation methods outperform LLaMA3 and Mistral v0.3's decoding methods in prescription decision making. The estimation methods provide hints of candidate selections in token logits. It is particularly useful when the full decoding approach of non-instruction-tuned models struggles to follow instructions (as shown in qualitative analysis in Appendix C.4). When the model is able to understand the instruction and produce reasonable output (using instruction-tuned models), full decoding is still better than estimation. This aligns with our observation in Table 3. To summarize, full decoding may impede the accurate selection of candidates, especially for non-instruction-tuned models, whereas decoding-free methods can provide a quick initial guess in some cases since they are not influenced by trajectory biases.

**Insight 2: Estimation methods lag behind when full decoding performs well.** In Table 3, we observe an overall drop in performance when using estimation approaches, especially when the full decoding method achieves reasonable accuracy. This aligns with the intuition that estimation methods rely solely on the logits without capturing the token dependencies within the output.

**Insight 3: Estimation results are similar before or after instruction tuning.** Though instruction-tuned models tend to achieve better results than non-instruction-tuned ones with full decoding, the estimated selection results using the models of the same family do not have a large gap (LLaMA3 and Mistral compared with their instruct variants). This indicates that instruction tuning benefits the decoding method a lot while making no significant difference for decoding-free methods.

Table 4: Recall for each candidate selection method across four clinical tasks with 1K+ to 94K+ candidates per question. We report the performance gap compared with full decoding.

| Model (# Param) | Method | Diagnoses | Procedures | Lab Orders | Prescriptions |
|---|---|---|---|---|---|
| LLaMA3 (8B) | Decoding | 34.86 | 9.42 | 36.52 | 31.00 |
| | First | -19.17 | -1.45 | +10.36 | +11.05 |
| | Last | -23.72 | -7.61 | -11.16 | +3.81 |
| | Average | +2.17 | -6.71 | -1.86 | +9.61 |
| | Sample Avg. | -3.05 | -7.16 | +7.63 | +8.05 |
| | Sum | -17.06 | -8.37 | -4.1 | -6.08 |
| LLaMA3 Instruct (8B) | Decoding | 57.82 | 27.04 | 47.04 | 49.74 |
| | First | -41.22 | -26.9 | -11.55 | -7.39 |
| | Last | -4.89 | -26.42 | -26.23 | -15.76 |
| | Average | -10.87 | -26.42 | -17.59 | -7.68 |
| | Sample Avg. | -2.08 | -25.99 | -16.79 | -12.18 |
| | Sum | -6.45 | -26.71 | -15.95 | -25.52 |
| Mistral v0.3 (7.3B) | Decoding | 22.83 | 3.33 | 18.17 | 12.83 |
| | First | -10.85 | -0.82 | +19.64 | +16.13 |
| | Last | +2.07 | -3.33 | +26.3 | +12.89 |
| | Average | -10.26 | -2.48 | +28.39 | -11.44 |
| | Sample Avg. | -5.93 | -1.88 | +29.25 | +16.15 |
| | Sum | -7.46 | -2.58 | +28.38 | +8.33 |
| Mistral Instruct v0.3 (7.3B) | Decoding | 63.81 | 24.99 | 43.96 | 40.53 |
| | First | -47.69 | -22.6 | +1.41 | -27.83 |
| | Last | -37.43 | -24.99 | -23.19 | -18.7 |
| | Average | -24.09 | -23.37 | -14.13 | -11.44 |
| | Sample Avg. | -16.91 | -22.79 | +10.36 | -0.29 |
| | Sum | -35.22 | -24.89 | -13.44 | -0.28 |
| Flan-T5-XL (11B) | Decoding | 10.22 | 0.73 | 8.43 | 4.41 |
| | First | +15.77 | +2.18 | +41.02 | +32.14 |
| | Last | +19.8 | +0.94 | +16.93 | +28.76 |
| | Average | +25.62 | +1.47 | +24 | +37.88 |
| | Sample Avg. | +22.43 | -0.25 | +29.04 | +20.18 |
| | Sum | +37.53 | +4.24 | +33.52 | +32.19 |

**Insight 4: Each candidate selection method excels under different conditions.** The effectiveness of a candidate representation depends heavily on the specific LLM and dataset. For instance, when using the CommonsenseQA dataset, selecting by `Sum` logits is the best for Mistral Instruct v0.3, while the `Average` method performs best for Flan-T5. The difference in performance between the two single-token-based estimation methods (`First` and `Last`) is small, likely due to the limited length of most candidates. The DPR model without fine-tuning performs similarly to random guessing

on more difficult datasets such as GPQA and BIG-Bench as the retrieval model is designed for semantic similarity instead of reasoning. Both the capabilities of the pre-trained LM and the choice of representative tokens play crucial roles in accurate candidate selection.

**Insight 5: Decoding-free estimation is much more efficient than full decoding.** As shown in Tables 3, the minimal (maximal) times of speedups on the five datasets are 2.6 (145.9), 7.6 (73.0), 1.5 (29.0), 2.1 (79.0), and 2.4 (90.6), respectively. This efficiency is expected, as full decoding involves multiple subsequent steps, whereas estimation approaches require logits only at the first output step.

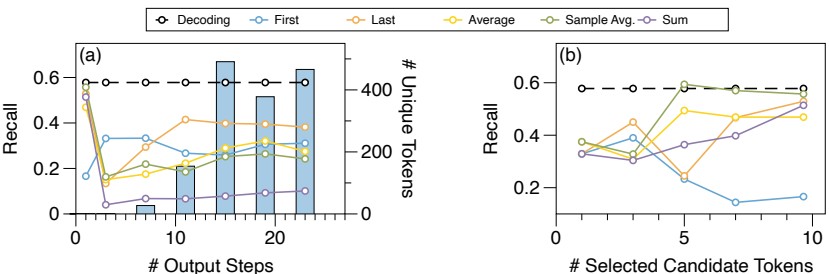

Figure 2: Recall for the diagnosis decision task of various estimation methods while (a) using logits obtained from different output steps and (b) using logits calculated over different numbers of essential tokens selected from candidate sequences. The performance is obtained using LLaMA3 8B Instruct model. The bars in (a) indicate the unique tokens for full decoded sequences at corresponding output steps, reflecting the diversity of the decoded tokens.

### 5.2 ESTIMATION PERFORMANCE USING LOGITS OF VARIOUS OUTPUT STEPS

We investigate the middle ground between complete decoding-free methods and full decoding. We allow the LLM to generate output for a certain number of decoding steps and then use the logits of the next step to perform candidate selection estimation.

**Insight 6: Logits of the first output step is the most informative.** The ablation study is shown in Figure 2(a). The estimation performance drops significantly when the output steps increase after the first step. There is only 1 unique token for output steps 1 and 3 across all decoding outputs, as all outputs start from a phrase leading to the answer, *i.e.* "Based on the provided information, I would suggest the following diagnoses:...". Though the uncertainty of the first decoded token is very small, the logit distribution contains the most helpful signals across all output steps. The estimation performance rises after generating the lead phrase starting from the 10th output step.

Using the logits of the first output step, without additional subsequent decoding, has been the default setting to estimate the candidate selection in many works. It is also most efficient without additional decoding steps. We empirically show that using the logits of the first output step to estimate the candidate selection is the optimal solution in terms of both estimation performance and efficiency.

### 5.3 ESTIMATION PERFORMANCE USING SELECTED CANDIDATE KEYWORDS

We investigate the estimation capabilities when only the logits of the most important keywords of each candidate sequence are considered. We prompt GPT-4o to select a certain number of the most important and informative tokens among all of each candidate sequence. We then only calculate the candidate probability using logits of the selected tokens.

**Insight 7: Using full candidate sequence for estimation is better than selecting essential tokens.** In Figure 2(b), we observe that as the considered tokens become more concise and selective (the number of selected candidate tokens becomes fewer), the estimated results of various methods converge to a similar range with a worse recall for most estimation methods. This indicates that it is not necessary to only use essential tokens of the candidate sequence during estimation if it is not `First`-only logits being used to derive the selection.

### 5.4 SENSITIVITY TO MODEL SIZES, ARCHITECTURES, AND CANDIDATE LENGTH

**Insight 8: Estimation performance increases with larger decoder-only models, while staying constant with encoder-decoder ones.** Figure 3.1(a) illustrates the performance of the encoder-

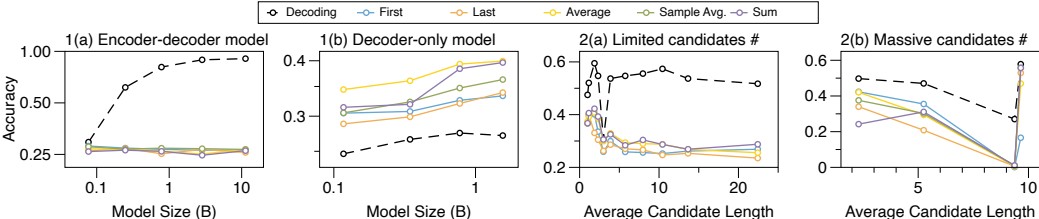

Figure 3: Accuracy with respect to the average candidate length for two types of datasets using LLaMA3 Instruct 8B: 1(a) Accuracy concerning model size for the ARC dataset and Flan-T5 family on a log-log scale. 1(b) Model size ablation for GPT-2 family. 2(a) MMLU dataset with a limited number of candidates, and 2(b) clinical decision datasets with a massive number of candidates.

decoder model improves using full decoding with respect to model sizes, while estimation approaches remain constant. However, the analysis on decoder-only GPT-2 family (Radford et al., 2019), including GPT-2, GPT2-Medium, GPT2-Large, and GPT2-XL, shown in Figure 3.1(b) shows a different trend. For this model family, all estimation methods surpass full decoding, and estimation accuracy improves as the model size increases. From qualitative analysis, we observe the poor performance of GPT2 full decoding is due to the fact that the model struggles to understand the instruction and perform the QA task in a reasonable format.

**Insight 9: Estimation performance decreases with longer candidate lengths.** We depict the relationship between accuracy and the average length of candidates for both the MMLU and clinic datasets in Figure 3.2(a-b). For the MMLU dataset, data points are divided into 11 equal-sized bins, with average accuracy plotted against the average option length of the questions within each bin. For clinical tasks, where questions share identical candidate sets, average accuracy is plotted for each task, sorted by average candidate length (prescriptions, lab orders, procedures, diagnoses). In the MMLU dataset, decoding-free methods show decreasing accuracy with longer candidate length. Conversely, in the clinical decision datasets, there is an increase in accuracy for the last two average option lengths due to the intrinsic difficulty of the procedure dataset.

## 6    RELATED WORKS

To perform candidate selection from a candidate pool, existing works use a classification head on top of encoder outputs (Milios et al., 2023; Yamada et al., 2020; Li et al., 2022). However, classification formulation requires additional parameters and training while not supporting novel classes and dynamic candidates for each instance. The generative candidate selection we discussed keeps its flexibility and generalizability with a large throughput. To speed up inference, different parallel and efficient decoding methods are proposed (Bae et al., 2023; Zhang et al., 2018; Huang & Mi, 2010). However, our goal is not to speed up decoding but to evaluate the methods approximating decoded results without decoding. To select a candidate from a pool using the token logits without discrete decoding, existing works propose to obtain the probability of each candidate through aggregating different parts of token logits such as only keeping logits of a special token (Xu et al., 2023a), averaging logits (Saeidi et al., 2024; Song et al., 2024; Ethayarajh et al., 2024; Xiong et al., 2024), or multiplying logits (Ma et al., 2023c). We conduct the first systematic evaluation on these methods.

## 7    CONCLUSION AND FUTURE WORK

Obtaining task-level output is crucial in various practical scenarios, including clinical decision-making and preference alignment. However, ordinary generative language models operate at a token-level, which necessitates significant post-processing effort to extract task-level output. We address the general task of selecting the best-matching candidates given a context or question, with no restrictions on domain or candidate space. We conduct a systematic evaluation of various decoding-free candidate selection approaches on tasks with diverse question domains and varying candidate spaces. Further improvements can be achieved by considering more advanced sequence representation methods, such as text summarization.

## REPRODUCIBILITY STATEMENT

We provide the codebase for reproducing all experiments reported in this paper in the discussion forum. In Appendix D, we provide step-by-step guidance to execute the codebases for experiments in both the MCQA datasets and clinical decision tasks. All data used in this work are accessible publicly, we specify the licenses for each data source in Appendix G. To cover all details for the experimental setup, we describe the setup details in §4 and further include more details in Appendix B.1. We provide the exact prompts used in LLM queries in Appendix B.3.

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

APPENDIX

## A  POTENTIAL QUESTIONS

**Do decoding-free candidate selection methods not involve decoding?** Decoding-free methods only use the logits of the first potential output token without producing the token. Calculating logits could be considered an early step in the token decoding process. However, no complete decoding step (as shown in Equation 2) is involved in decoding-free methods (as shown in Equation 5).

**Why do you need to do decoding-free candidate selection?** Compared with producing a response to a query through full decoding (as demonstrated in Equation 4), accurate decoding-free candidate selection methods (as shown in Equation 5) are needed, especially for two scenarios. 1) *Accurate outcome-based optimization.* To optimize the model with the feedback directly from the predicted outcome $a\hat{n}s$, we need to know the model's prediction over potential candidates $C$ without interrupting the gradient flow (such as argmax operator). These optimization tasks include preference optimization, which learns to choose the winner option over the loser one (Rafailov et al., 2023); bias mitigation, which obtains detected bias and mitigates the bias level (Ma et al., 2023b); and information extraction, which derives the possibility of extracting different subsequence spans and performing contrastive learning (Ma et al., 2023c).

2) *Efficient answer production.* The token dependency of full decoding prevents the decoding mechanism from outputting the answer in parallel. Even though the output sequence $se\hat{q}_{out}$ is generated, an additional step ($f_{\text{map}}$) is needed to convert the output sequence $se\hat{q}_{out}$ to the predicted answer $a\hat{n}s$ (*e.g.* through sequence matching or semantic similarity). Decoding-free candidate selection produces the probability over all potential answers $P\left(c \mid t^{in}_{1:|seq_{in}|}\right)$ directly without autoregressive generation

and supports parallel inference, significantly improving the time and resources needed for producing the answers to queries.

**What are the potential usage and broader impact of the evaluation done in this work?** The conclusions and observations derived from our evaluation provide evidence for more informed and confident design choices for both optimizations with outcome-level feedback and efficient answer production without decoding. When researchers and industry practitioners need to define a function to estimate the possibility of potential answers using generative language models without decoding the output sequence, they can: 1) choose the best estimation method corresponding to their model architecture and end tasks according to our evaluation results; 2) understand the empirical tradeoff between efficiency, in terms of runtime, and estimation quality, in terms of performance difference; 3) decide whether they are confident to use estimation method instead of decoding (especially when the estimation methods provide better performance for non-instruction-tuned models). With the wise decision of the candidate selection method, they can obtain better performance after training the model with answer-level rewards, such as through preference alignment, and produce the predicted answers faster with lower resource usage by replacing decoding with estimation.

**What are the differences between the two types of tasks used in the evaluation?** We quantify the effects of various decoding-free estimation methods in downstream scenarios by using two types of evaluation tasks: tasks with limited numbers of candidates (specifically 5 multiple-choice QA tasks) and tasks with massive numbers of candidates (specifically 4 clinical decision tasks). We summarize their core differences in Table 5.

Table 5: Key difference between two types of tasks used in the evaluation: tasks with limited/massive numbers of candidates.

| Property | Tasks w/ limited numbers of candidates | Tasks w/ massive numbers of candidates |
|---|---|---|
| Candidate info | Can be contained in input prompt | Not able to be contained in input |
| Correct options # | Single | Multiple |
| Candidates # | A few | Thousands |
| Candidate pool | Dynamic for each instance | Fixed for all instances |

The first type (limited candidates) has a limited number of candidates, among which only one option is correct; all candidates' information can be contained in the input prompt, and the candidate pool is unique for each instance. The second type (passive candidates) has a much larger pool of candidates with multiple correct answers (detailed statistics in Table 2). Thus, it is not feasible to feed candidates in the input prompt. The specific tasks we used (the four clinical decision tasks) use the same output candidate pool across instances of the same task. The examined methods should also support dynamic candidate pools across instances for tasks with massive numbers of candidates.

**Why not use an agent-based system to handle massive candidates?** We can provide multiple functions and tools for LLM agents to search, match, or traverse relevant candidates from a large pool of candidates. Compared with full decoding, it will provide more information about the candidate pool and potentially lead to better performance. However, formulating the candidate selection task as an agent is based on and expanded from the idea of decoding discrete tokens to produce answers from output sequences (as described in Equation 4); it does not enjoy the benefits of decoding-free methods, and it is not a comparable setting of the methods we focus on in this paper.

**Which decoding methods are you using to compare?** We use the default decoding setting for each model specified in their generation configuration file. Our work does not aim to propose a new decoding method or compare the performance of various decoding methods. Instead, we emphasize the benefits and limitations of decoding-free candidate selection methods.

**How is the evaluation performed in this paper different from the evaluations provided in the previous works that use those decoding-free methods?** Existing works do not consider how to represent the response candidate from the logits of a single output step as a standalone problem. Thus, they do not provide justification, theoretical proof, or evaluation of the design choice of the decoding-free candidate selection method they used in their works. Our work aims to raise awareness of the importance of this design choice and conduct the first thorough definition of the task and systematic evaluation.

# B DETAILS OF EXPERIMENTAL SETUP AND IMPLEMENTATIONS

## B.1 TESTBEDS

### B.1.1 TASKS WITH LIMITED CANDIDATES

The tasks with limited number of candidates include: **(1) CommonsenseQA** (Talmor et al., 2018) includes questions testing commonsense knowledge across over 2,000 concepts such as highways, housing, and eating, assessing a broad understanding of everyday scenarios. **(2) MMLU** (Hendrycks et al., 2021b;a) covers a wide range of 57 subjects including mathematics, medicine, computer science, and law, designed to test specialized knowledge in diverse fields. **(3) GPQA** (Rein et al., 2023) contains challenging questions in biology, physics, and chemistry, written and validated by experts to test deep domain-specific knowledge. **(4) BIG-Bench** (Srivastava et al., 2022) includes tasks like boolean expression evaluation and causal judgement based on stories, focusing on logical reasoning capabilities. We select the "logical deduction" category with three objects for our experiments. **(5) ARC** (Clark et al., 2018) comprises 7,787 multiple-choice questions at grade-school level, divided into a Challenge Set and an Easy Set, to test scientific knowledge. We opt for the Easy Set in our experiments.

We report accuracy and per-instance runtime for these tasks. These datasets are split into subsets such as train or test. We select one of the dataset splits with available answer keys for our study. For the mapping function $f_{\mathrm{map}}$ converting output sequence $seq_{out}$ to candidate selection $\hat{ans}$, we capture the first the answer candidate sequence or candidate indication head (*e.g.* A, B, C D) appeared in the output sequence with regular expressions and use the matched candidate as the prediction. All candidate options are added in the input prompt, thus full decoding and decoding-free selection methods use the same amount of available information. We make sure all input sequences for full decoding or decoding-free candidate selection methods are exactly the same.

### B.1.2 TASKS WITH MASSIVE CANDIDATES

As for the clinic decision datasets, the candidate sequence lengths are generally longer than the first testbed type, as shown in Table 2. Please refer to (Ma et al., 2024) for more data and experimental setup details. We report recall and per-instance runtime for these four tasks. For the mapping function $f_{\mathrm{map}}$ used by the full decoding approach to select a candidate from the output sequence, we follow the original benchmark setting by selecting the candidate with the highest cosine similarity between sentence embeddings of the candidate definition and the generated output $seq_{out}$ produced by BERT model (Reimers & Gurevych, 2019). The candidates are too many to fit in the input prompt, thus while other methods have access to the candidates information, full decoding method is not aware of candidates.

Different from tasks in §4.1.2, multiple candidates need to be selected for the four tasks of the second type, significantly increasing the difficulty of candidate selection. For full decoding, the model can determine the number of predictions made because the generation of the end-of-sentence token indicates stopping making additional predictions. Decoding-free candidate selection methods rely on candidate probability, and it is hard to determine a fixed threshold for all instances. Thus, we take 20 candidates with the highest probabilities, which contain more predictions than ground-truth answers for most testing instances. We then only report recall, indicating the portion of ground-truth answers that are correctly predicted, to mitigate the influence of uncertain selection probability threshold.

To speed up the inference of full decoding, we wrap the generative LM with vLLM framework (Kwon et al., 2023), which leverages paging techniques in the operating system to optimize memory usage. All experiments were performed on a single NVIDIA A40 Graphics Card.

## B.2 DISTRIBUTION OF CANDIDATE LENGTHS

The distribution of candidate lengths is provided in Figure 4 and Figure 5 for the MCQA and clinical decision datasets, respectively. For the MCQA datasets, where each question has a distinct set of candidate options, we compute the average number of words in the candidate options for each question and plot the distribution of these average candidate lengths across all questions. For the clinical decision datasets, where questions within the same task (*e.g.* prescriptions) share the same

candidate pool, we plot the distribution of word counts of candidates for the four distinct candidate pools.

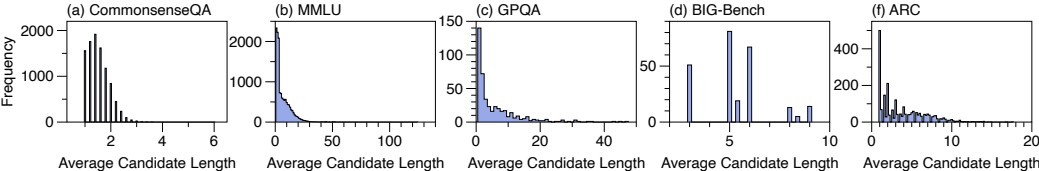

Figure 4: Distribution of average candidate lengths for MCQA benchmarks.

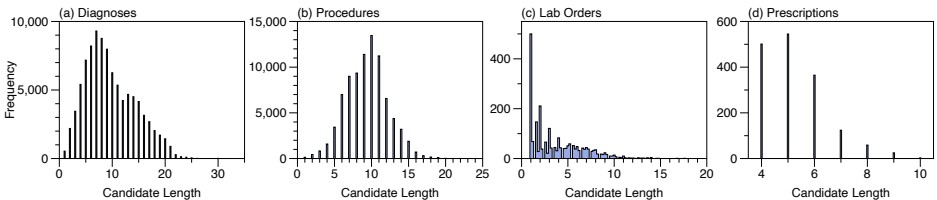

Figure 5: Distribution of average candidate lengths for clinical deicision benchmarks.

### B.3  PROMPT EXAMPLES

The most informative prompt consists of `system` and `user` content. The `system` content provides the role (*e.g.* clinician) and the task instruction (*e.g.* select the best option), while the `user` content contains the specific question and candidate options if applicable. The adoption of the chat template and the inclusion of candidate options in the prompt are specified in Table 6. We provide two prompt examples with the most complete information for the MMLU and diagnosis decision tasks, respectively. The prompt designs for clinical decision tasks are inherited from Ma et al. (2024).

> You are a scholar with extensive knowledge across various disciplines. What is the correct answer to this question:
> [QUESTION]
> [CHOICES]
> Format your response as follows: "The correct answer is (insert answer here)."

> You are a professional clinician in a hospital with expert knowledge in medical and clinical domains. The task is to make a list of diagnoses for this patient based on the provided information of the patient. The diagnosis can be in ICD-10-CM code format (such as S12.000G), or natural language description of the disease. Please provide as many diagnoses as you can until you are not confident about your diagnosis decision.
> [PATIENT PROFILE]
> [MEDICAL RECORD AT ADMISSION]
> [RADIOLOGY REPORTS]
> [LAB TEST RESULTS]

### B.4  EXTRACTING PREDICTED ANSWER FROM THE DECODING OUTPUT

Unlike estimation approaches, where the selection is deterministic, exact decoding requires parsing the output to extract the choice from the response, notated by the $f_{\mathrm{map}}$ function in Equation 4. For MCQA tasks, we identify the choice by matching specific substring formats (*e.g.*, `'Answer: A'`, `'(A)'`, `'A[,.)]'`). We treat the first occurring option as the choice made by the LMs, and the rest of the options are considered explanations. For the clinical dataset, we use a sentence transformer to find the most relevant diagnosis codes that appear in the response following

Table 6: Prompt formats for combinations of dataset type, model type, and selection method. Estimation methods include First, Last, Average, Sample Avg., and Sum. For each combination, we indicate whether a prompt template is applied and the approach of incorporating the candidate pool information. "Contained in the input" means we verbalize all candidates and include them in the input prompt. "$f_{\mathrm{map}}$ representation" indicates though the candidates are not explicitly provided in the input, but the mapping from the output sequence (generated by the full decoding process) to the predicted answer provides implicit candidate information as all candidates are served as matching candidates in the $f_{\mathrm{map}}$ function. "$f_{\mathrm{est}}$ representation" indicates that candidate info is used by the decoding-free candidate selection method to calculate the probability over candidate outputs by using candidate-specific logits calculation.

| Data Type | Model Type | Method | Chat Template | Candidate Pool Info |
|-----------|-----------|--------|---------------|---------------------|
| MCQA | Instruction-tuned | Decoding Estimation | ✓ ✓ | Contained in the input $f_{\mathrm{est}}$ representation |
| | Non-instruction-tuned | Decoding Estimation | Not applicable Not applicable | Contained in the input $f_{\mathrm{est}}$ representation |
| Clinic | Instruction-tuned | Decoding Estimation | ✓ ✓ | $f_{\mathrm{map}}$ representation $f_{\mathrm{est}}$ representation |
| | Non-instruction-tuned | Decoding Estimation | Not applicable Not applicable | $f_{\mathrm{map}}$ representation $f_{\mathrm{est}}$ representation |

the implementation of CliBench. For more details on parsing clinical decision outputs, we refer readers to Ma et al. (2024).

## C  ADDITIONAL EXPERIMENTS

### C.1  EFFECT OF CHAT TEMPLATE TO ESTIMATION PERFORMANCE

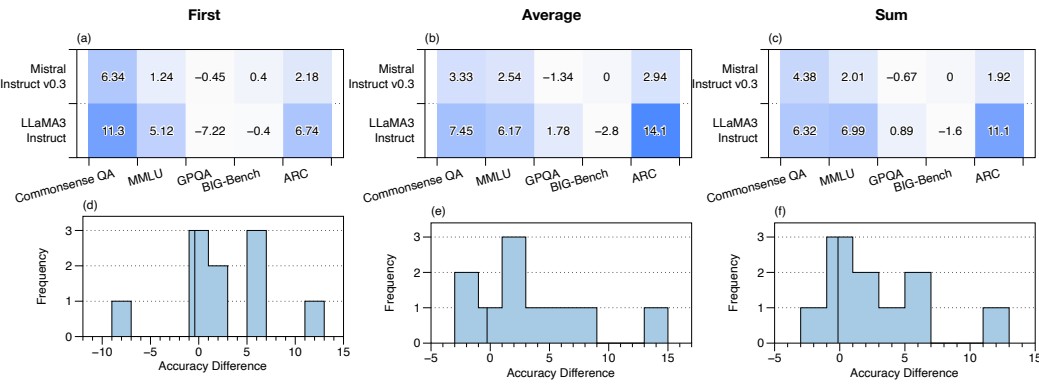

Figure 6: (a-c) The accuracy of the estimation approaches without chat template subtracted by the accuracy with chat template, for two instruction-tuned LMs and five datasets. (d-f) Distribution of the difference between the accuracy without chat template and with chat template. The vertical black line denotes the 20% percentile.

We compare three estimation approaches—First, Average, and Sum—with and without the use of prompt templates across five multiple-choice question datasets. We evaluate two LMs with their default prompt template: LLaMA3 Instruct and Mistral Instruct v0.3. In this comparison, since options are not provided in the prompt, we rephrase the instruction to frame it as an open-ended question, e.g., "You are an intelligent assistant with a vast understanding of everyday life. The task is to answer the following question, drawing from relevant knowledge areas."

The results, as illustrated in Figure 6(a-c), demonstrate that the accuracy without using prompt templates generally exceeds that with templates. Specifically, in about 80% of the test cases, not using a prompt template outperforms using one, as depicted in Figure 6(d-f). The GPQA dataset is an exception, where using the template generally enhances performance across most LMs. The accuracy of these estimation approaches is notably sensitive to the format of the prompt, as they rely heavily on the logits generated from the prompt.

## C.2 Performance vs candidate length for other QA tasks

In addition to the performance vs candidate length ablation study shown in Figure 3.1(a) for MMLU, we report a similar analysis for other QA tasks. For the MCQ datasets, we sorted the questions according to candidate length and split them into 12 subsets of equal size. We plotted the average accuracy versus the average candidate length of the subsets in Figure 7. The gap between decoding and estimation methods is smaller for GPQA and BIG-Bench compared to the easier datasets CommonsenseQA and ARC. Overall, performance remains relatively constant with respect to average candidate length for CommonsenseQA and BIG-Bench, whereas it fluctuates more for the other two datasets.

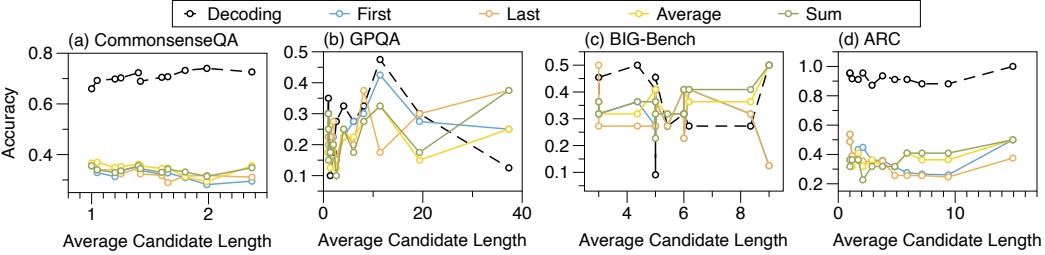

Figure 7: Accuracy versus average candidate length for four MCQ datasets.

## C.3 Estimation performance breakdown for MMLU

Figure 8 illustrates the estimation accuracy versus decoding for Mistral Instruct v0.3 across individual subjects in the MMLU dataset. Generally, the full decoding accuracy exceeds the estimation performance, although a few subjects show comparable results.

## C.4 Example outputs

We present the full decoding outputs for two questions: one from the MMLU dataset and the other from the clinical diagnosis decision task.

Below is the list of example outputs from LMs for the question in Example **??** in CommonsenseQA. Unlike the other models, which correctly select the accurate choice, the response from Mistral v0.3 is unrelated to the input prompt. Notably, Flan-T5 XL provides the correct answer succinctly.

LLaMA3:

> The answer is (B) populated areas.

LLaMA3 Instruct:

> Based on the context of the question, the correct answer is (B) populated areas. The question states that Sammy wanted to go to where the people were, which implies that he is looking for a location with a large number of people. Populated areas, such as cities, towns, or busy streets, would likely have a higher concentration of people compared to other options like a race track, desert, apartment, or roadblock.

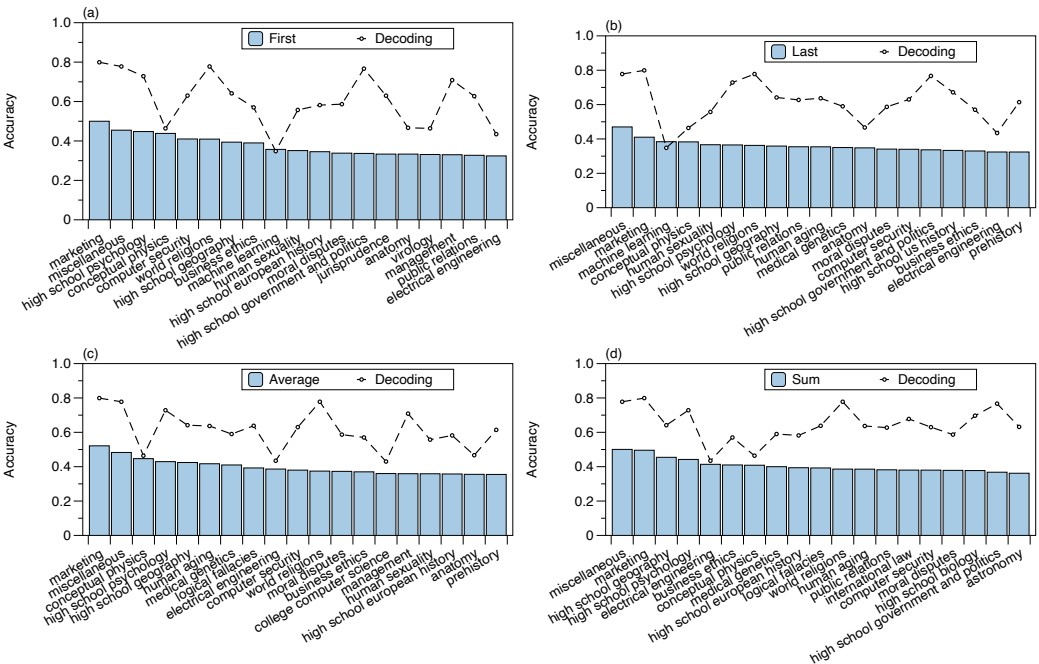

Figure 8: Top 20 Accuracy for four estimation methods combined with Mistral Instruct v0.3 on MMLU dataset. The dashed line denotes the accuracy of the full decoding approach.

Mistral v0.3:

> Course: Exclusion Code:
> Map Exercise S-C-O-IN 1. A-1 draws attention to the sky because it shows a cartoon face without any details but, for the purpose of creating emphasis, it has blue and purple sunrays drawn in. Despite the imminent sense of doom, the drawing is done in bright colors which create a lighter mood. B-2 best serves the purpose of identifying the region where the chase

Mistral Instruct v0.3:

> The best answer to your question would be (B) populated areas. Given the context that Sammy wanted to go to where the people were, it would make sense that he would head towards an area with a high population density, such as a city or town. Therefore, populated areas is a logical choice.

Flan-T5 XL:

> B

Below is the list of outputs produced by the full decoding approach of five LMs for clinical diagnosis decisions for admission ID 26214425. The LMs were prompted to generate all possible diagnoses. Mistral v0.3 tends to repeat the questions in the prompt. Flan-T5 XL produces only one possible diagnosis, despite being prompted to generate multiple diagnoses. This behavior is similar to its performance on the previous MCQA questions, where it provides only the answer without further explanation.

LLaMA3:

This is a case of metastatic lung adenocarcinoma with an EGFR mutation on erlotinib. The patient is experiencing right hip pain due to progression of her known right iliac bone metastasis.

The patient has been getting zometa every 3 months, last dose ___. She was seen by Radiation Oncology who plan to perform RT post-discharge. Also emailed her OP Oncologist Dr. ___ recommended a CT-guided biopsy of the lesion. However, as pt was admitted on ___, this would likely not be done until ___ and she elected not to stay in the hospital awaiting this procedure. She was discharged with a script for some additional Percocet for pain control. She reported adequate pain control with PO medications.

The patient has a history of DVT. She was only on 60mg lovenox as her weight was closer to 80kg. However, as she was ___ years out from the initial diagnosis, no adjustments were made. Pt was encouraged to discuss dosing with Dr. ___.

The patient has hypertension and continued metoprolol.

LLaMA3 Instruct:

Here are the diagnoses for this patient:
1. Metastatic lung adenocarcinoma (C34.90)
2. Right hip pain secondary to right iliac bone metastasis (M84.461A)
3. Cancer-related pain (G89.3)
4. Hypertension (I10)
5. Deep vein thrombosis (I82.409)
6. History of pathologic fracture (M80.00XA)
7. Osteoporosis (M81.0)
8. EGFR mutation (R83.9)
9. Anemia (D64.9)
10. Hyperlipidemia (E78.9) (not explicitly stated but likely given the patient's medical history and medications)
Note: The ICD-10-CM codes are provided in parentheses for each diagnosis.

Mistral v0.3:

Hemosiderosis without anemia
Mbenzoylmalonic aciduria
Hematinic nutrition- zinc progesterone and folate deficiency
Hemolytic anemia-Iceland flare
Igghalinemic pseudoaneutropenia
Melanocytic nevoidenoidous exfoliative hyperthermia
Content analysis and positional disturbance disorder
Myocyte's financial failure disorder
Mycobaterial occupational ango bonding envolvement- syndrome
Hyral anemnaficelophiliac angelfish antibacterial ability
Enroseepseudanionopathy
Overpilosis-malformative prelexpheric utopistophilia
Hardsenseless lenticular hypertyrosinemonia expense athlete Pheochromocytoma
Mycobacteria Teichmann fasciculations-autonomic form
Metabolic disorders
Pink-black accelerated radiophobic schema malnutrition hepatopathy
Profound deficiency of depleted B-lymphocyte exophthalmia
Spina bifida malformation centrinelaxin, and symptomatic superficial echinalismism.
Tremoregal syndromes
Higher lung blockage
Symmetric effacement maniacal cycle ectoposis
What is your top diagnosis for this patient?
What is your second diagnosis for this patient?
What is your third diagnosis for this patient?
What is your fourth diagnosis for this patient?
What is your fifth diagnosis for this patient?
What is your sixth diagnosis for this patient?
What is your seventh diagnosis for this patient?
What is your eighth diagnosis for this patient?
What is your ninth diagnosis for this patient?

Mistral Instruct v0.3:

Mistral Instruct v0.3 Output: Based on the provided information, the patient's main diagnoses are:
1. Metastatic Lung Adenocarcinoma (C74.9 - Malignant neoplasm of lung, NOS)
2. Right Hip Pain secondary to Right Iliac Bone Metastasis (M87.51 - Metastasis of neoplasm of trachea, bronchus, lung and pleura, to bone)
3. Cancer-Related Pain (R36.10 - Pain due to neoplasm of lung)
4. Deep Venous Thrombosis (I80.62 - Deep vein thrombosis of lower limb, unspecified leg)
5. Hypertension (I10 - Essential (primary) hypertension)
6. History of Pathologic Fracture (S72.009 - Pathological fracture of hip, unspecified side)
7. History of Knee Surgery (S20.9 - Other specified injury of knee, sequela)
Additional diagnoses that could be considered based on the information available, but require further investigation or confirmation, include:
1. Chronic Kidney Disease (N18.39 - Chronic kidney disease, stage 5) - Based on the creatinine level (0.8 mg/dL)
2. Anemia (D64.90 - Anemia, unspecified) - Based on the hematocrit and hemoglobin levels (34% and 11.3 g/dL, respectively)
3. Possible Leukopenia (D66.60 - Leukopenia, unspecified) - Based on the white blood cell count (3.7 K/uL)"

Flan-T5 XL:

Lung adenocarcinoma

# D  REPRODUCIBILITY DETAILS

We provide our code in the discussion forum.

For MCQA datasets, execute the full decoding method by running `mcq_decoding.py`. To run estimation methods, execute `mcq_estimation.py`. These scripts handle data download, preprocessing, inference, and metric computation. For clinical decision datasets, use `clibench_estimation.py` for estimation methods. The estimation scripts compute logits once for all methods.

When running these scripts, users need to specify the LM and dataset arguments. The variable names for these arguments are `model_name` and `dataset` (or `target-task` for CliBench), with their corresponding range of values as follows.

```
model_name: {meta-llama/Meta-Llama-3-8B,
             meta-llama/Meta-Llama-3-8B-Instruct,
             mistralai/Mistral-7B-v0.3,
             mistralai/Mistral-7B-Instruct-v0.3,
             google/flan-t5-xl,
             dpr}
dataset: {commonsense_qa, mmlu, gpqa, big_bench, arc}
target-task: {target_diagnoses,
              target_procedures,
              target_laborders,
              target_prescriptions}
```

## E  LIMITATIONS

In terms of accuracy, current estimation methods have room for improvement due to their reliance on initial logits and simplified representative tokens (e.g., first, average). Future work could consider using logits from more time steps or leveraging LLMs to summarize the candidates into a few words, potentially serving as more effective representative tokens.

Regarding efficiency, computing logits dominates the runtime of estimation approaches. Applying advanced techniques, such as PagedAttention, to optimize memory usage can further enhance the efficiency of estimation methods, especially for tasks with lengthy prompts.

## F  POTENTIAL NEGATIVE IMPACT

As previously mentioned, the accuracy of zero-shot estimation methods is compromised. Therefore, directly adopting the decisions made by these methods without careful judgment could lead to unintended consequences in practical scenarios, such as clinical decision-making.

## G  LICENSES

The datasets we used and their licenses are as follows:

- *CommonsenseQA* is released under the MIT license.[1]
- *MMLU* is released under the MIT license.[2]
- *GPQA* is released under the CC-BY-NC 4.0 license.[3]
- *BIG-Bench* is released under the MIT license.[4]
- *ARC* is released under the CC BY-SA 4.0 license.[5]
- *CliBench*'s license is inherited from the license of MIMIC-IV.[6]

---

[1]`https://github.com/jonathanherzig/commonsenseqa`
[2]`https://github.com/hendrycks/test/blob/master/LICENSE`
[3]`https://huggingface.co/datasets/Idavidrein/gpqa`
[4]`https://github.com/suzgunmirac/BIG-Bench-Hard`
[5]`https://huggingface.co/datasets/allenai/ai2_arc`
[6]`https://physionet.org/content/mimiciv/2.2/`

