# OpenReview forum: "Inferring from Logits: Exploring Best Practices for Decoding-Free Generative Candidate Selection"
_ICLR.cc/2025/Conference — Submitted to ICLR 2025_

### Official Review · Reviewer_n7sb · 2024-11-01

**Soundness:** 2
**Presentation:** 2
**Contribution:** 2
**Rating:** 3
**Confidence:** 4

**Summary:**

This paper evaluates different approaches for candidate selection using large language models (LLMs). Specifically, it compares two categories of methods: (i) decoding, where the model generates a candidate based on an instruction and query, followed by post-processing, and (ii) estimation, where candidate selection is based on examining the logits of the model in response to the instruction and query. The authors benchmark models of various sizes, both with and without instruction fine-tuning, across candidate selection tasks of varying difficulty. They find that estimation methods generally improve the performance of models without instruction fine-tuning but reduce the performance of instruction-fine-tuned models. Additionally, they observe that the effectiveness of estimation methods varies by task.

**Strengths:**

When considering how to perform candidate selection with generative models, there are several design choices to make: for example, should we generate candidates by decoding, or infer them directly from the model’s logits? Evaluating how these decisions affect downstream performance could be helpful for practitioners in making informed choices. For this reason, I find the premise of this paper quite valuable.

**Weaknesses:**

**Framing:** The framing in this paper, especially in the abstract, is unclear and could be improved. For instance, the abstract begins by discussing the limitations of decoding candidates from LLMs, which gives the impression that the paper will propose more efficient methods for candidate selection. To address this, I suggest introducing the evaluation focus right from the start to set clearer expectations.

**Presentation:** The paper’s presentation could be significantly improved in a few areas:

1. Consistency: Each category of methods should have a consistent label. Currently, terms like “decoding-free methods,” “estimation methods,” and “candidate selection without decoding” are all used to describe the same category, which can confuse readers.

2. Avoid vague terms: Certain terms and descriptions lack clarity. For example, the authors mention that “decoding-based methods cut off the gradient flow and prevent optimization,” without specifying what kind of gradient flow or optimization issue they’re referring to. In another instance (line 390), they state that “decoding-free methods are not influenced by trajectory biases,” but it’s unclear what “trajectory” means or what types of biases are involved. Similarly, in line 395, they write, “estimation methods rely solely on the logits without capturing the token dependencies within the output,” which could benefit from a more precise explanation.

**Method:** In an evaluation paper, one of the main objectives is to provide insights that practitioners can use. However, this paper lacks interpretations of the results, often leaving the reader with more questions than answers. For instance, in Table 4, it appears that estimation methods consistently reduce the performance of instruction-tuned models. One might expect that, since instruction-tuned models are more calibrated for task performance, their logits would align better with the reference candidates. However, there’s a large performance gap between estimation and decoding methods, with estimation methods performing notably worse. Additionally, the table shows variations in the performance of different estimation techniques across tasks, but the authors do not analyze or interpret these differences, only noting that certain techniques work better for some tasks than others. Finally, an essential aspect missing from the evaluation is the best practice of performing calibration [1] during decoding, which is not addressed in the paper.

[1] [Calibrate Before Use: Improving Few-Shot Performance of Language Models, Zhao et al.](https://arxiv.org/pdf/2102.09690)

**Questions:**

Most questions are in the weakness section.

Line 462: why isn't that phrase part of the prompt? usually, the first token refers to the first token in the answer, right?

---

### Official Review · Reviewer_TyT1 · 2024-11-04

**Soundness:** 2
**Presentation:** 2
**Contribution:** 2
**Rating:** 3
**Confidence:** 4

**Summary:**

Various methods exist to parse task predictions from large language model (LLM) outputs, yet they lack systematic definition, evaluation, and comparison. Addressing this gap, the authors systematically evaluate decoding-free generative candidate selection methods for LLMs, focusing on scenarios where task-level predictions are made from a set of candidates. The evaluation covers five decoding-free methods alongside the other approaches, such as classification, retrieval, and full decoding, tested across five multiple-choice question (MCQ) datasets, like CommonsenseQA and MMLU, as well as four additional clinical decision-making tasks. From the experimental results, the authors conclude nine key insights to guide future development and application of LLMs.

**Strengths:**

1. The focus on decoding-free methods shows their potential to reduce computational costs, an increasingly important factor in deploying LLMs in real-time, resource-intensive environments.
2. The paper offers a systematic evaluation of existing candidate selection methods for LLMs, serving as a useful reference given the rapid development and expanding applications of LLMs.
3. The experiments include multiple MCQ datasets and model types, providing actionable insights into the strengths and limitations of each method and thus the practical utility of the findings.

**Weaknesses:**

1. Technically speaking, the paper functions more as an analysis, evaluating existing methods rather than proposing new approaches. While insights are drawn from the experiments, unfortunately, no new methods are proposed given the concluded insights. Thus, the lack of novel methodology may limit its contribution for a top-tier conference.
2. Although multiple datasets are included, the types of tasks remain limited. Standard classification tasks, such as those in the GLUE benchmark for evaluating natural language understanding, are missing. Instead, the choice of clinical decision tasks is not well-motivated or explained, which may lead to results that are biased toward a specific domain.
3. Although the paper provides insights from experimental results, it lacks a clear recommendation on the best technique to use in general settings. The insights largely focus on numerical comparisons without deeper investigation, leaving it unclear which method is generally preferable or in what contexts specific methods may excel, and the according reason.

**Questions:**

1. What is the definition of "decoding" in the context of "full decoding" in this paper? Are not the logits considered part of the decoding process? If so, what specifically makes logits-based methods "decoding-free"?
2. The table summarizing differences across various methods is helpful. Since the mapping function is task-dependent, it would be useful to this in the table.
3. The experimental setup and results are not organized in a standard format. It would improve clarity to separate different task groups into individual sections rather than combining all setups in one section and all results in another.

---

### Official Review · Reviewer_ma7Z · 2024-11-04

**Soundness:** 2
**Presentation:** 2
**Contribution:** 2
**Rating:** 3
**Confidence:** 4

**Summary:**

The paper explores how to use standard autoregressive language models for tasks that require, for a given query, selection from a candidate from predefined candidate pool (making this a form of classification problem). Instead of doing autoregressive decoding and then mapping the output sequence to a candidate, the authors explore “decoding-free” approaches directly select a candidate.

They evaluate several decoding-free methods, including using the logits from the first, last and all tokens from each candidate: multiple-choice QA with limited candidate pools and clinical decision-making with extensive candidate options. Their analysis spans various large language models, from non-instruction-tuned to instruction-tuned, and considers different architectures and model sizes.

The study finds that decoding-free methods *sometimes* are effective alternatives to full-decoding, especially when the model has relatively low performance with decoding-based selection. However, for most cases, it seems decoding an output sequence and matching it to a candidate seems to yield the best results.

**Strengths:**

- The problem explored is practically relevant: doing classification with decoded outputs from LLMs is always troublesome, requiring dealing with edge cases when they don’t decode one of the possible candidates, marginalization of different lexical forms for candidates, etc… Alternatives methods that perform as well or better would be very valuable.
- Some positive results for classification tasks with very large number of labels/candidates

**Weaknesses:**

- My main concern is the practical usefulness of the results: the methods explored seem to quite lacklustre in terms of performance compared normal decoding-based selection. This, coupled with the lack of novelty in proposed methodology (as this is mostly a comparison paper) leads me to not see much actionable insight to take away from this paper. I suggest the clarify and specify practical insights for readers to take away (which methods to use, in which case, etc…)
- The paper also doesn’t touch/compare a fundamental capability that decoding-based approaches allow: chain-of-though. How would the final performance and speed of decoding-based approaches change when chain-of-though is considered, and how does that change the picture of the comparison to decoding-free approaches?
- Overall the paper also has some presentation issues, with a somewhat confusing terminology in section 2 and some typos over the paper.

**Questions:**

- What are the speeds for the large candidate pool tasks? It seems that it is the case where decoding-free approaches could be more beneficial compared to decoding-based, but it’s unclear what the speed difference is here (as I expected in this case that decoding-free approaches are actually slower)

---

> ### Author Response · Authors · 2024-11-23
>
> **W1 Evaluation and analysis paper instead of method paper**
> - We would like to clarify that we aim to provide insights of existing decoding-free candidate selection methods, instead of proposing a method that outperforms the full decoding method.
> - We echo your point that our paper is a comparison paper instead of trying to propose a new method for decoding-free candidate selection.
>
> **W1 Poor performance of the decoding-free methods**
> - Decoding-free and full decoding methods are two paradigms, and thus cannot be directly replaced interchangeably. The less ideal performance of decoding-free method is under the strong condition that they are much more efficient than full decoding methods.
> - Having better performance using decoding-free methods compared with full decoding methods is neither our claim nor our expectation of this paper. Our goal is to provide the problem formulation and testbeds to evaluate the characteristics of decoding-free methods. We are not proposing a new method that outperforms full decoding.
>
> **W1 Actionable/practical insight**
> - The actionable suggestions can be straightforwardly derived from the evaluation results demonstrated in the insights points in Section 5, including:
>   - Consider using decoding-free methods when the tasks are challenging and full decoding struggles to verbalize the results.
>   - When efficiency is the target, use decoding-free methods to find the sweet spot between performance vs efficiency.
>   - Use the logits of the first output step (do not use the middle ground method to get logits after a few output steps) when estimating the candidate selection using decoding-free methods.
>   - Use the full candidate sequence is the best practice instead of summarizing or selecting essential tokens as concise candidates
>   - Use a larger size decoder-only model can improve the estimation performance, while using a larger encoder-decoder model does not lead to better performance
>   - Use decoding-free methods with caution if the candidate sequences are very long
>
> **W2 Performance with reasoning steps**
> - Setting: we evaluate the estimation performance using CoT reasoning by using the following three settings on MMLU college_mathematics using Llama-3-8B-Instruct:
>   - No CoT, logits of 1st output step: the default setting used in the submission paper
>   - CoT, logits of 1st output step: the input prompt contains CoT instruction and in-context examples. We use the logits of the first output step to feed in the decoding-free estimation methods
>   - CoT, logits after thoughts: similar as above, but we let the model decode and generate the step-by-step thinking process, and then we take the logits of the output step right before generating the final answer
> - Results: the results are shown in the table below (value indicates the performance difference compared to full decoding). We observe:
>   - CoT did improve the estimation performance
>   - Explicitly including the generated thoughts in the condition of the logits used for estimation degrades the performance. This aligns with our “Insight 6” in Section 5.2 where we show the logits of the first step are the most informative
> - We will include a more extensive analysis regarding CoT’s effect on decoding-free estimation methods in our revision.
>
> ||No CoT, 1st step|CoT, 1st step|CoT, after thoughts|
> |-|-|-|-|
> |First|-9|-7|-11|
> |Last|-5|-4|-10|
> |Sample avg|-6|-4|-10|
> |Avg|-7|-6|-7|
> |Sum|-2|0|-5|
>
> **Q1 Speeds for the large candidate pool tasks**
> - We show the runtime averaged per instance for all methods on some clinical tasks in the table below. We will also make sure to include this data in our revision.
> - Decoding-free approaches are 30x faster than full decoding, which aligns with your expectation that decoding-free is more beneficial when the candidate pool is larger.
>
> |Task|Method|Runtime (s) |
> |--|--|--|
> | Diagnoses| Full Decoding|424.16|
> | | First|14.48|
> | | Last| 14.10 |
> | | Avg| 15.67|
> | | Sample Avg. | 15.18|
> || Sum| 15.59|
> | Lab Orders | Full Decoding |491.98|
> | | First| 3.60|
> | | Last| 2.99|
> | | Avg | 3.06 |
> | | Sample Avg. | 2.97|
> | | Sum| 3.07 |
>
> **W3 Presentation issues**
> - Thanks a lot for your suggestions on the presentation and notations. During writing, we try to align our terming and equations to the ones used in the book “Speech and Language Processing”. In the first question in Appendix A, we also try to clarify further the concept “decoding” used in our context in case that’s the concern.
> - Please let us know if you have any specific notations or terminologies that might be confusing, and we will make sure to improve those writings.
> - We will screen the entire paper and fix the typos in our revision.
>
> We are encouraged to see you recognize the problem we analyzed is practically relevant. We hope the additional experiments and responses we provided will be helpful. Please let us know if there are any other concerns that we might be able to address. Thanks a lot!

---

### Official Review · Reviewer_c2d6 · 2024-11-09

**Soundness:** 3
**Presentation:** 2
**Contribution:** 2
**Rating:** 3
**Confidence:** 4

**Summary:**

This paper proposes a method for selecting an answer among candidates from a language model without explicitly decoding the answer. Given a question and candidates, a typical scheme would use them as prompt and generate an answer from the LM. In case of extreme classification in which the number of candidates is numerous, the candidates might be omitted from the prompt. The generated answer is then compared against the candidates for selection. However, this process is expensive as it requires a full decoding pass. This paper proposes to use the logits produced directly after encoding the question (and candidates) to induce a distribution over the candidates and pick the highest scoring candidates. This approach doesn't need to generate the full output sequence which could be faster in the scenario when the candidates are long and numerous. Various approaches for using the logits produced after encoding are tested. The experiments are carried out on multiple choice selection tasks and extreme classification tasks with a very large number of lengthy candidates.

**Strengths:**

-- The proposed problem is well motivated. Not having to decode the answer in mutiple choice QA has potential to imporve runtimes significantly.

-- The proposed approach is simple to implement and is faster tham retrieval and full-decoding based approaches.

-- The experiments on base-versions of language models seem to be promising for short candidate answers.

**Weaknesses:**

-- My biggest concern is that the proposed approach shows very poor results in general. For the case in which candidates are not numerous and can be incorprated into the prompt (Table 3), the proposed approach singificantly degrades the performance over nearly all the datasets and LMs. While it shows improvements over the base LMs, it is handily outperformed by instuction-tuned versions of these models.

-- Similar concerns are observed on tasks in which candidates are numerous and long (Table 4). Based on these numbers, I am not convinced that logits produced after encoding contain enough information to help with classification effectively. In table 4, the approach does imporve over Flan-T5. But this model is a very poor baseline to begin with compared to the instruct-LMs.

-- Moreover the performance of the proposed approach degrades with longer candidate lengths. This is precisely the setting in which one would want to use a decoding-free candidate selection approach to save on potentially large decoding costs.

-- The writing could be made mroe concise and direct to describe the ideas in the paper.

**Questions:**

Please address the concerns above

---

> ### Author Response · Authors · 2024-11-22
>
> Thanks a lot for your review!
>
> We would like to clarify that we aim to provide insights of existing decoding-free candidate selection methods, instead of proposing a method that outperforms the full decoding method.
>
> **Poor results for the “proposed” approach**
> - Decoding-free and full decoding methods are two paradigms, and thus cannot be directly replaced interchangeably. The less ideal performance of decoding-free method is under the strong condition that they are much more efficient than full decoding methods.
> - Having better performance using decoding-free method compared with full decoding methods is neither our claim nor our expectation of this paper. Our goal is to provide the problem formulation and testbeds to evaluate characteristics of decoding-free methods. We are not proposing a new method outperforming full decoding.
>
> **Whether logits produced after encoding contain enough information to help with classification**
> - We are not claiming that logits contain enough information, rather we are exactly trying to investigate the open questions
>   - Whether logits contains sufficient information?
>   - How to induce those information?
>   - What scenarios this information can be helpful?
> - The insights induced from evaluation results (Section 5.1) also indicate it’s not a question with a black-or-white answer. That’s exactly why it is worth a paper to investigate. We are glad to see the questions you raised align with our motivation.
>
> **Performance of the “proposed” approach degrades with longer candidate lengths**
> - We present this insight with grounded evaluation results with empirical evidence: the decoding-free candidate selection methods can estimate the candidate selection with decreased performance when the candidate lengths are getting longer.
> - This insight provides clear guidance of future method proposers and model developers to use decoding-free methods with caution when candidate sequences are long.
> - This is a property we induced from our quantitative analysis. As we are not proposing a better decoding-free method for long candidate scenarios. It’s hard for us to consider this insight as a weakness. We are glad to see you find this insight interesting.
>
> **Writing can direct to describe the ideas**
> - We didn’t directly describe the ideas because we are not proposing a method. Instead, we introduce the task formulation, formal definition, existing paradigms and insights of the properties of these decoding-free candidate selection methods. To make the task formulation clear, we spend some paragraphs introducing existing setting and notations.
> - We will re-organize our writing to make it more concise.
>
> We are excited to learn that you recognized our problem well motivated, the approach is simple and faster and the experiments are promising for short candidates.
>
> Please let us know if there are any other concerns or questions we could address! We hope you could reconsider the rating given our clarification and further information provided in the rebuttal.
>
> Thanks a lot!

---

> > ### Comment · Reviewer_c2d6 · 2024-11-26
> > **Thanks for the rebuttal**
> >
> > Hello, thank you for your response. My impression of the paper remains the same so I am leaving my score unchanged.

---

### Meta-Review · Area_Chair_BE8c · 2024-12-22

**Metareview:**

The paper addresses an important area of using LLMs for candidate selection and presents a systematic evaluation of different methods. However, the concerns raised by the reviewers regarding performance compared to existing baselines, lack of novelty in methodology, and presentation issues are valid. While the authors have provided reasonable responses and clarifications, it remains to be seen if these adequately address the core concerns to a level that would change the overall perception of the paper's suitability for acceptance.
Considering the multiple rejections from the reviewers, this paper should be further improved and considered to submit to another avenue.

**Additional Comments On Reviewer Discussion:**

Multiple reviewers pointed out that the decoding-free approach shows relatively poor performance compared to full decoding methods in many cases and its performance degrades with longer candidate lengths. They questioned the practical usefulness of the results given this performance gap. The authors clarified that they are not aiming to propose a method that outperforms full decoding but rather to provide insights into existing decoding-free candidate selection methods. They presented evaluation results as properties and insights, such as when to use decoding-free methods considering efficiency and task difficulty, and how different settings impact performance (e.g., using logits of the first output step, considering candidate sequence length, etc.). Additionally, the authors emphasized their goal of evaluating and providing a comprehensive understanding of the characteristics of decoding-free methods, which they believe is valuable in the context of the rapid development and application of LLMs.

---

### Decision · Program_Chairs · 2025-01-22

Reject